# Progressive Classifier Mechanism for Bridge Expansion Joint Health Status Monitoring System Based on Acoustic Sensors

**DOI:** 10.3390/s23115090

**Published:** 2023-05-26

**Authors:** Xulong Zhang, Zihao Cheng, Li Du, Yuan Du

**Affiliations:** 1School of Electronic Science and Engineering, Nanjing University, Nanjing 210023, China; xl_zhang@smail.nju.edu.cn; 2School of Automation, Nanjing University of Information Science and Technology, Nanjing 210044, China; zihao_cheng@outlook.com

**Keywords:** IoT, acoustic sensor, fault diagnose and classification, end-to-cloud coordinated

## Abstract

The application of IoT (Internet of Things) technology to the health monitoring of expansion joints is of great importance in enhancing the efficiency of bridge expansion joint maintenance. In this study, a low-power, high-efficiency, end-to-cloud coordinated monitoring system analyzes acoustic signals to identify faults in bridge expansion joints. To address the issue of scarce authentic data related to bridge expansion joint failures, an expansion joint damage simulation data collection platform is established for well-annotated datasets. Based on this, a progressive two-level classifier mechanism is proposed, combining template matching based on AMPD (Automatic Peak Detection) and deep learning algorithms based on VMD (Variational Mode Decomposition), denoising, and utilizing edge and cloud computing power efficiently. The simulation-based datasets were used to test the two-level algorithm, with the first-level edge-end template matching algorithm achieving fault detection rates of 93.3% and the second-level cloud-based deep learning algorithm achieving classification accuracy of 98.4%. The proposed system in this paper has demonstrated efficient performance in monitoring the health of expansion joints, according to the aforementioned results.

## 1. Introduction

Over the past few years, the development of transportation infrastructure has led to the construction of a significant number of bridges. As a result, ensuring bridge safety, improving operation efficiency, and reducing maintenance costs are of substantial social and economic significance. For this reason, both government officials and transportation enterprises have paid close attention to these objectives. Bridge expansion joints, in particular, are highly susceptible to damage caused by construction defects, vehicle overloading, and other issues that arise during bridge maintenance activities. Consequently, the effective monitoring of the health status of bridge expansion joints has become an urgent concern.

Given the recent advancements in computing hardware, Internet of Things devices have broad application prospects in structural health monitoring. IoT devices can be efficiently deployed and utilized for monitoring the health status of these structures through the use of edge computing technology, allowing for terminal-cloud collaboration and storage-calculation integration, which ensures the safe operation of bridges while reducing the consumption of resources.

Data acquisition modules for structural health monitoring traditionally rely on various sensors, including stress sensors [1], acceleration sensors [2], acoustic sensors [3], fiber optic sensors [4,5], and other sensors. These sensors provide parameters such as vibration frequency and modal vibration of the bridge structure, enabling the identification of different health conditions. However, each of these sensors has its own drawbacks, such as being susceptible to environmental interference, difficult to annotate data, and expensive to deploy on a large scale [6,7]. Experienced engineers usually rely on the sound of passing vehicles to locate faults in bridge expansion joint maintenance. As a result, the system uses acoustic sensors as data collectors to identify the health status of the bridge. This allows us to collaborate with experienced engineers to annotate large-scale data accurately.

Various techniques have been developed and implemented in structural health monitoring (SHM), including both local and data-based methods. Local techniques are primarily focused on specific sensor locations and their responses to structural changes. Meanwhile, data-based methods rely on a larger group of sensors distributed throughout the structure to collect generalized information on the state of the structure. While both approaches have their strengths, there are trade-offs to consider. For example, local techniques are highly sensitive to local damage but may produce false alarms depending on the location of the installed sensors. Data-based methods, while more robust in detecting global damage, can be less accurate at identifying localized damage [7,8].

The sound signal is essentially a vibration signal. Presently, a significant amount of research has been devoted to applying vibration signals in the development of structural health monitoring algorithms [9]. Traditional signal processing encompasses methods, such as kurtosis, peak-value factor, and mean root square, to process time-domain signals, while fast Fourier transform and envelope spectrum transform are used to analyze signals in the frequency-domain. Nevertheless, these processing techniques are limited by their susceptibility to environmental factors and poor robustness. The current mainstream approach is to employ modern signal processing methods to perform an in-depth analysis of original signals to extract features. Commonly used modern techniques include time-frequency analysis and modal decomposition.

Time-frequency analysis is a vital tool for signal processing, which can simultaneously display the characteristics of the time domain and frequency domain on the same image. Several methods widely used for time-frequency analysis are the discrete wavelet transform (DWT), short-time Fourier transform (STFT), and the dual-tree complex wavelet transform algorithm based on these methods [10]. These algorithms facilitate signal denoising and feature extraction, among other subsequent operations. Therefore, they are useful in various applications, especially in fault diagnosis [11,12].

Mode decomposition algorithms are essentially nonlinear methods for analyzing signal frequency-intensity distributions. These include the Empirical Mode Decomposition (EMD) algorithm proposed by Huang et al. [13] in 1998, the Ensemble Empirical Mode Decomposition (EEMD) algorithm introduced by Huang et al. in 2009, which alleviates mode-mixing effects by adding white noise [14], and the Complete Ensemble Empirical Mode Decomposition with Adaptive Noise (CEEMDAN) algorithm proposed by Torres M.E. et al. in 2011, which adds noise to the decomposition process for processing [15]. Previous studies [16,17] have explored the use of these methods for feature extraction and classification. However, the EMD algorithm’s lack of mathematical interpretability, mode-mixing effects, and endpoint effects, along with its high time and space complexity, pose significant challenges.

In 2014, Dragomiretskiy et al. [18] proposed a new adaptive signal decomposition method, Variational Mode Decomposition (VMD), which utilizes strict variation equations to derive optimal solutions yielding a good modal decomposition effect and a strict theoretical underpinning. The research conducted by Marco Civera and colleagues shows that the VMD algorithm outperforms the CEEMDAN algorithm in the field of structural health monitoring (SHM) when the signal is highly noisy. Additionally, VMD also exhibits better frequency shift tracking ability during signal decomposition. Moreover, VMD takes nearly 1/20th of the computation time compared to CEEMDAN in processing the same signal [19]. Based on its theoretical underpinning and advantages mentioned above, VMD will be considered for further processing of signals in this study.

Data-driven methods based on machine learning and deep learning models are increasingly used in equipment fault detection due to advancements in hardware computing power. A plethora of conventional algorithms, such as Support Vector Machine (SVM), Random Forest, Convolutional Neural Network (CNN), Long Short-Term Memory (LSTM), and others [16,17,20,21], have been employed in fault diagnosis. Moreover, the integration of biologically-inspired optimization techniques, such as Genetic Algorithm (GA), Particle Swarm Optimization (PSO), and Grey Wolf Optimization (GWO), has been pursued to enhance the precision of fault diagnosis and prediction [22,23,24,25,26].

Li et al. proposed a new method for identifying multiple parameters of concrete dams via an integration of polynomial chaos expansion and slime mold algorithm. Unlike other optimization techniques, this method preserves a high level of precision while reducing the computational burden. Nevertheless, the accuracy of this method relies on the norm selection and degree of the polynomial employed [25]. H. Tran-Ngoc et al. introduced an ANN-PSO-GA optimization algorithm to enhance the accuracy and reduce the computational complexity of artificial neural networks. By contrast to the conventional PSO method, this combined optimization mechanism yields better precision and lower computational demands. Notwithstanding, the authors acknowledged that while ANN requires fewer computational resources, its ability to extract features falls short of deep learning approaches, such as CNN [26].

Nevertheless, the potential of these data-based algorithms is constrained by the capability of the device and the size of the dataset. Although they have vast potential for application, they are not mature yet.

The direct application of modern signal processing, machine learning, parameter optimization, and other related algorithms to edge computing puts high demands on the computing power and power consumption of edge devices. As a solution, this paper utilizes the progressive algorithm design concept and bifurcates the algorithm into two levels. During normal times of the expansion joint, low-computing-demand algorithms are executed to minimize the high load on edge device hardware resources. Edge computing can execute these algorithms solely based on its computing power. It can efficiently monitor devices’ health status while reducing data transmissions. In cases of an abnormal expansion joint, complex algorithms such as modern signal processing will denoise and extract features from signals, ultimately providing accurate judgment and classification. This strategy can effectively resolve the issue of prolonged processing time and high computing power, and resource demands due to complex data processing.

This paper makes the following contributions:A low-power and high-efficiency edge computing system is designed in this paper for “edge-cloud collaboration” and “storage-computation integration”. The system is capable of monitoring bridge expansion joints’ health status periodically.A platform was developed to simulate and collect data on expansion joint damage. The platform was augmented with an edge data collection device to capture a substantial amount of accurately annotated data, which was appropriate for developing subsequent algorithms.A two-level classifier mechanism is proposed in this paper, which is particularly suitable for edge computing. The proposed mechanism is combined with template matching algorithms, which run exclusively on edge devices, and neural network models with different computing power requirements that reduce the computing power demands in practical applications and improve the monitoring accuracy and efficiency.

## 2. Progressive Two-Level Fault Diagnosis Algorithm

### 2.1. Overview

The initial step in diagnosing faults in expansion joints based on sound data is to extract valid segments of the raw data. Then these extracted segments are input into a two-level classifier. In the first level, a template-based algorithm with low computational consumption is utilized for performing preliminary diagnosis. The algorithm involves Fast Fourier Transform (FFT) and Automatic Peak Detection (AMPD). This algorithm is suitable for use on edge computing devices. The second level of classification employs a CNN based on VMD for accurate diagnosis. As shown in Figure 1.

### 2.2. Slice (Extract Valid Segments)

The device provides 30 s of raw data, and the sample rate is *Fs*, yielding 30 × *Fs* data points. Valid segments refer to sound segments captured when vehicles pass over the expansion joint, which has been tested to last about half a second. Thus, the algorithm extracts a valid segment containing 1 s of sound data consisting of Fs sample points.

The algorithm establishes three sliding windows to traverse the entire raw data, including a main window of length *Fs* and two subordinative windows, each of length *Fs*/4. The maximum amplitude value of each sliding window is computed. The algorithm considers a segment valid only if the maximum amplitude value of the main window is at least six times greater than the maximum value of either one of the subordinative windows and selects the main window as a valid segment in such instance.

However, the maximum amplitude value of the selected segment may not lie at the window’s center, which may cause disruption to subsequent algorithms. Consequently, we adopt an iterative approach whereby we use two sliding windows of length *Fs*/2 positioned on either side of the maximum amplitude value to reposition the maximum value at the window’s center until the process converges.

The effect of the slice algorithm is shown in Figure 2.

### 2.3. Automatic Peak Detection (AMPD)

Felix Scholkmann et al. [27] proposed the Automatic Multiscale Peak Detection (AMPD) method for detecting peaks in periodic noise signals and quasiperiodic signals. However, the algorithm mentioned in the literature is computationally expensive in terms of time and space complexity, prompting us to simplify it to some extent. Results from experiments have verified that the simplified algorithm performs well in detecting peaks and envelopes for non-periodic signals.

Let S be a univariate signal sampled uniformly and represented as $\left[s_{1},s_{2},\ldots ,s_{Fs}\right]$. The simplified steps of the AMPD algorithm based on the multiscale approach are as follows:

Step 1: Calculate the Local Maximum Magnitude Matrix (LMM). Firstly, determine the local maximum scale of the signal *S* using the sliding window method. Assign *k* to range between 1 and *Fs*/2 + 1 and *i* to range between *k* and *Fs/2* − *k* for the sliding window. When the following conditions are met:(1)$$s_{i}>s_{i-k}\;\mathrm{and}\;s_{i}>s_{i+k}$$

The corresponding scale $w_{k}$ expands accordingly with the sliding. After the sliding process, the scale matrix $W=[w_{1},w_{2},\ldots ,w_{\mathrm{Fs}/2+1}]$ is obtained.

The method of acquiring the maximum scale image is simplified, reducing the space complexity of the original algorithm from O (n^2^) to O (n), although this may come at the expense of the algorithm’s effectiveness in certain exceptional situations. This simplification eliminates the least squares fitting process and reduces the time required for operation.

Step 2: Determine the maximum window length, $L_{\max}$, by obtaining the value of k that corresponds to the maximum scale on the scale matrix.

Step 3: Scan the entire dataset by sliding a window obtained in Step 2. When $s_{i}$ represents the maximum value of a $L_{\max}$ length window, a peak is identified. Obtaining all peaks of the original signal can be achieved by following the above method.

For a periodic signal, the peak interval remains fixed, whereas, for a non-periodic signal, this algorithm is capable of effectively capturing the peaks of each interval. Refer to the illustration below for more clarity.

The result of AMPD is shown in Figure 3.

### 2.4. Template Matching-Based Determination Algorithm (First Level)

The algorithm based on templates is similar to human intuition. A basic template is provided for comparison by analyzing the frequency domain curve of valid segments. This enables us to distinguish the type of valid segment. The specific process is as follows:

Step 1: Perform FFT on the time series *S* of the valid 1 s segment, transform the time domain signal with weakly rule-based to the frequency domain (taking only the first half of the FFT result due to symmetry) using a sampling rate of *Fs*.
(2)$$S_{fft}=\mathrm{FFT}(S,Fs)$$
(3)$$S_{fft\_\log}=20\times \log_{10}S_{fft}$$

Step 2: Utilize the AMPD algorithm to extract the peak values from frequency domain sequences, and the upper contour is obtained as *Peaks*.
(4)$$Peaks=AMPD(S_{\mathrm{fft}\_\log})$$

Step 3: The signal in the frequency domain is partitioned into 20 identical intervals. The representative value of each interval is determined by selecting its maximum peak value. These 20 maximum peak values constitute the higher contour of the signal.
(5)$$env=[peak_{\max1},peak_{\max2},\ldots ,peak_{\max20}]$$

Step 4: Select *N* valid 1 s segments (usually equal to the number of segments in 2–3 days) and repeat steps 1–3 above to obtain the upper contour $env_{i}$. Based on the upper contour matrix Env of the *N* segments, choose the maximum and minimum values at each point, and extend them outward by specific values $tol_{up}$ and $tol_{down}$, respectively. This produces two boundaries of the defined region, the upper boundary $Region_{up}$ and the lower boundary $Region_{down}$.
(6)$$Env=[env_{1};env_{2};\ldots ;env_{20}]$$
(7)$$\begin{array}{l} Region_{up}=env_{\max}+tol_{up}\\ Region_{down}=env_{\min}+tol_{down} \end{array}$$

Step 5: For a valid 1 s segment, steps 1–3 are taken to obtain its upper contour $env_{test}$. These 20 data points are then compared with a pre-defined region. If more than 10 points lie outside the region, the segment is judged as abnormal; 1–10 points suggest an anomaly in the segment, and all points within the region indicate the segment is normal.

Step 6: After each maintenance of the expansion joint, repeat steps 1–4 to update the template to account for any changes in the condition of the equipment and seek to enhance the system’s robustness while also minimizing the impact that equipment aging and updates to system components have on the accuracy of the algorithm.

### 2.5. CNN Classifier Based on VMD Denoising (Second Level)

The valid segments sent to the cloud are decomposed by VMD into several intrinsic mode functions (IMF) with different central frequencies. Then correlation coefficients are used to denoise. The reconstructed time series is then input into the CNN for fault diagnosis and classification.

#### 2.5.1. Variational Mode Decomposition (VMD)

VMD is a novel adaptive signal decomposition method that was proposed by Dragomiretskiy [16] in recent years. The method constructs a variation equation and searches for the optimal solution to that equation. The original signal is placed into the variation model. It then decomposes the original signal into adaptive components by solving the variation model. VMD is a fully non-recursive decomposition method, which effectively mitigates mode aliasing and endpoint effects that may appear during the decomposition of signals through EMD. VMD decomposes complex signals into numerous analysis signals with unique central frequencies and sparsity features using Wiener filtering and Hilbert transform. It also computes the marginal spectrum of each analysis signal. Afterward, VMD shifts the central frequency of each mode using the displacement property of the Fourier transform through multiplication by $e^{jwkt}$, demodulates each mode’s spectrum to the baseband, and estimates the broadband using $L^{2}$ norm. The final variation constraint model is presented as the VMD equation:(8)$$\left\{\begin{array}{c} \underset{\left\{u_{k}\right\},\left\{\omega_{k}\right\}}{\min}\left\{\left.\overset{K}{\underset{k=1}{\sum }}\right\|{\left.\partial_{t}\left[\left(\delta \left(t\right)+\frac{j}{\pi t}\right)*u_{k}\left(t\right)\right]e^{-j\omega_{k}t}\right\|}_{2}^{2}\right\}\\ s.t.\overset{K}{\underset{k=1}{\sum }}u_{k}=f\left(t\right) \end{array}\right.$$

In the above constraint model, *f* represents the input signal, *t* represents time, δ(t) represents the Dirac distribution, ∗ represents convolution operation, and $\left\{u_{k}\}=\{u_{1},u_{2},\ldots ,u_{K}\right\}$ represents intrinsic mode functions (IMF). $\left\{\omega_{k}\}=\{\omega_{1},\omega_{2},\ldots ,\omega_{K}\right\}$ represents the central frequency of each IMF. In order to facilitate the calculation and ensure the absolute integrability of each component, a quadratic penalty factor-alpha, and a Lagrange multiplier factor λ(t) are introduced to obtain an extended Lagrange expression. The variational constraint problem is then transformed into an unconstrained problem.
(9)$$\begin{array}{ll} L\left(\left\{u_{k}\right\},\left\{\omega_{k}\right\},\lambda \right)= & \alpha \left.\overset{K}{\underset{k=1}{\sum }}\right\|{\left.\partial_{t}\left[\left(\delta \left(t\right)+\frac{j}{\pi t}\right)*u_{k}\left(t\right)\right]e^{-j\omega_{k}t}\right\|}_{2}^{2}\\ & +\left\|f\left(t\right)\right.-\overset{K}{\underset{k=1}{\sum }}{\left.u_{k}\left(t\right)\right\|}_{2}^{2}\\ & +<\lambda (t),f(t)-\overset{K}{\underset{k=1}{\sum }}u_{k}\left(t\right)> \end{array}$$
where α is a secondary penalty factor and λ is the Lagrange operator. By using the alternating direction multiplier algorithm, the saddle point of the Lagrange function is obtained, which is the optimal solution of the constrained variational model, with the mode component $u_{k}$ and the central frequency $\omega_{k}$.

In this model, *K* is set at 8, the penalty factor alpha is 7000, and a maximum of 500 iterations are performed. After VMD, our valid data segments are decomposed into eight modes, $imf_{1}$ to $imf_{8}$, and are arranged in ascending order of their central frequency. The decomposition result is shown in Figure 4.

#### 2.5.2. Mode Reservation Based on Pearson Correlation Coefficient

The Pearson correlation coefficient is computed between the eight mode components and the original signal. The formula for calculating the Pearson correlation coefficient of a single IMF component is given below.
(10)$$r=\frac{\sum_{i=1}^{n}\left(S_{i}-\bar{S})(IMF_{i}-\bar{IMF}\right)}{\sqrt{\sum_{i=1}^{n}{\left(S_{i}-\bar{S}\right)}^{2}}\sqrt{\sum_{i=1}^{n}{\left(IMF_{i}-\bar{IMF}\right)}^{2}}}$$
where *S* represents the original signal, $\bar{S}$ represents its mean value, *IMF* represents a mode, and $\bar{IMF}$ represents its mean value.

The degree of relatedness between each IMF component and the original signal is obtained by calculating the formula. The correlation between the IMF component and the original signal is greater when $\left|r\right|$ is closer to 1. Three IMF components with the smallest correlation coefficients are removed, and noise attenuation is completed, followed by feature enhancement. Figure 5 illustrates the denoising effect.

#### 2.5.3. CNN Classification

The convolutional neural network (CNN) is a type of artificial neural network that has been widely applied in various fields, such as speech recognition and computer vision. It has greatly contributed to significant advancements in these fields. CNN is usually implemented using the popular and powerful framework BVLC Caffe [28,29] from the University of California, Berkeley.

CNN has three principal layers:the Convolutional Layer, which is responsible for extracting features;the Max Pooling Layer, which down-samples input data and reduces data volume without impairing classification results.the Fully Connected Layer, which is responsible for classification tasks.

In this study, each valid segment S after denoising and FFT is reconstructed as a square matrix $S_{r\_fft}$, which was subsequently utilized as the input of CNN. The CNN network we built has the following topology, which is shown in Figure 6:The first convolutional layer has a 1D input and a 5 × 5 kernel, producing 16D output channels. Additionally, it has a padding size of 2 and a stride of 1.The second convolutional layer has a 16D input and a 5 × 5 kernel, producing 32D output channels. It also has a padding size of 2 and a stride of 1.A fully connected layer with 32,768 units.A dropout layer to prevent overfitting during the training process.A fully connected layer with 128 units.Finally, a fully connected layer with 3 units is employed for classification.

To accelerate convergence learning and introduce nonlinearity into the proposed system, ReLU (Rectified Linear Unit) layer is applied to all convolutional and fully connected layers, which sets all negative activation values of a given input to zero by utilizing the following function: f(x)−max(0,x).

After each ReLU layer, a Max Pooling layer (down-sampling layer) with a 2 × 2 filter is applied to decrease the spatial dimension.

## 3. Periodical Monitoring System

### 3.1. Overview

The health monitoring system for bridge expansion joints comprises two components: cloud server and edge data acquisition and processing devices, as illustrated in Figure 7. Further, edge data acquisition and processing devices can be subdivided into three modules: data acquisition, processing, and transmission.

### 3.2. Cloud Server

In this section, we utilize Tencent Cloud for managing the devices. The cloud service platform is responsible for managing the devices online, which includes tasks such as timing startup, switching the devices to engineering mode, updating parameters such as sampling rate, etc. Furthermore, some software algorithms are deployed on the cloud server. During the system runtime, the collected sound data and diagnostic results are saved.

### 3.3. Edge Data Collection and Processing Device

MEMS microphones are installed in edge devices under the expansion joints of the bridge to regularly collect 30 s sound data. The Microcontroller Unit (MCU) in the edge device processes the data, such as extracting effective segments and executing first-level algorithms.

Through site, investigations and sound signal analysis using high-performance equipment such as phones, normal and abnormal expansion joints have a characteristic frequency in the lower frequency band. Specifically, features within the band of 3 kHz are sufficient for detecting faults. As a result, a sampling rate of 8 kHz is applied for the bridge expansion joint health monitoring system. However, during subsequent iterations, the sampling rate can be manually adjusted to meet the needs of different types of expansion joints and faults.

Given the network conditions at the bridge, as well as the desire for efficient data transmission and low power consumption, the 4G-Cat1 module with an extended antenna is chosen for data transmission. The module connects to the MCU using a pin header, enabling inter-board communication through the serial port protocol. The MCU uses AT commands to control Cat1. Results are sent to a remote server using the HTTP protocol. If necessary, data are returned to the cloud server for accurate classification using the second-level algorithm. Moreover, the data transmission module receives data, such as switch signals, between engineering and normal operation modes and server settings, which satisfies the need to set specific parameters for different scenarios.

The edge device is shown in Figure 8.

### 3.4. System Operation Process

The process is shown in Figure 9. Procedures carried out by the edge device are represented in the blue box, while those executed by the cloud server are represented in the yellow box. A detailed explanation of the procedures inside the white box will be provided in the subsequent text.

#### 3.4.1. Device Boot Initialization

The edge devices located at the bridge expansion joint are scheduled to wake up and receive instructions from the server. The devices check whether the firmware version stored in the firmware update folder on the server matches that inside the device. If firmware versions do not match, the updated firmware code is sent to the device using the HTTP protocol and remotely updated via the bootloader, which is deployed in the MCU. After updating, the device reads the server-issued parameter settings, which include the boot-up interval, sampling time, and sampling rate, and completes the initialization process. If there is a match, the edge device reads the parameter settings and performs subsequent operations, including data collection and processing.

#### 3.4.2. Data Acquisition and Processing

After the initialization of the equipment, the device enters the acquisition mode and sets the necessary parameters for data collection. Then, the device conducts valid segment extraction. These extracted valid segments are then sequentially input into the first-level algorithm that is deployed in the MCU. Once the first-level algorithm extracts the features, it checks for the existence of the corresponding template file. If the template file does not exist, it is created and initialized. If it already exists, the number that determines whether to update the template and the template interval is read. The device then checks whether it is in engineering mode. If it is indeed in engineering mode, the default result output is set to normal, and the template is reinitialized. However, if it is not in engineering mode, the number is compared against the preset threshold. If the number is greater than the threshold, the health status judgment is conducted. Conversely, if the number is less than the threshold, the feature of this segment is used to update the template, and the default output result is set to normal.

#### 3.4.3. Data Transmission and Diagnose

If the running of the first-level algorithm results in abnormal or suspicious outcomes, the second-level algorithm must run. Conversely, if normal outcomes are acquired, the server only receives diagnostic results, which inform the expansion joint condition of the maintenance personnel responsible for the bridge.

Once the first-level algorithm has finished processing, the cloud server acquires the running results of this algorithm and assesses the flag indicating whether the second-level algorithm is running. If the second-level algorithm needs to execute, the valid segments are reserved and dispatched to the data transmission module. Utilizing the CNN model deployed on the server, the received data are categorized and diagnosed more meticulously, culminating in the output of diagnostic results.

#### 3.4.4. Enter the Low-Power Sleep Mode

After the device completes the above three steps, it transitions into a low-power mode, where it awaits awakening for the next operational cycle.

After measuring, an edge device only consumes 2 mAh of power during the activation process while carrying out procedures such as data collection, preliminary assessment, and data transmission. Notably, the current in low-power mode is only a few microamps. With six daily activations, its energy consumption reaches 16 mAh. Equipped with a 40,000 mAh battery, the device would be expected to operate for two or more years while taking into account aging and voltage factors affecting its power supply.

### 3.5. Expansion Joint Damage Simulation Data Collection Platform

Bridge expansion joint failure data collection is quite challenging, and labeling is an issue. Currently, there is no available dataset regarding sound segments of joint expansion failures. The study aims to create a well-labeled and balanced dataset for algorithmic research by building a simple expansion joint damage simulation data collection platform. For this purpose, the platform is composed of two crosswise connected aluminum alloy structures and rubber between two types of connecting pieces that simulate the rigid and flexible components of real expansion joints. The experiment utilized a spinning top car and a scale to ensure consistency in the speed of the car during the simulation. As shown in Figure 10.

Different failure situations can be observed by adjusting the fixed position of the connecting piece, and this flexibility allows researchers to gather the required data. This platform provides a trustworthy source of data for testing expansion joint failure diagnosis algorithms.

## 4. Results

### 4.1. Overview of Dataset

The main data used in the algorithm proposed in this article are provided by the bridge expansion joint damage simulation platform mentioned in the third section. Here, we simulated three situations: normal, main connection loose (MCL), and subordinate connection loose (SCL). Normally, the screws between the primary connection plates and the main body of the analog bridge, including the screws between the two types of connection plates, are tightly fastened. The term “main connection loose” signifies the screws becoming loose or dislodged between the main connection plate and the bridge’s bodies, while “subordinate connection loose” refers to the screws becoming loose or dislodged between the main connection plate and the subordinate connection plate.

We collected 500 sets of effective fragments for the verification of the fault diagnosis algorithm, as shown in Table 1. The training set and test set are divided in a ratio of 7:3.

The comparison of the time-domain and frequency-domain of the collected three signals is shown in Figure 11 and Figure 12.

A comparison of the three signals shows differences both in the time domain and frequency domain. In particular, there are obvious differences between the normal segments and the faulty segments. Thus, applying a template-matching algorithm for initial classification is considered a practical approach.

While the frequency domain distribution differences may not be as prominent for both types of faulty segments, their characteristic frequencies differ around 2–3 KHz after undergoing VMD, which is illustrated in Figure 13. Therefore, it is feasible to reconstruct the signal using VMD and then proceed with further classification.

### 4.2. Results Based on Template Matching-Based Determination Algorithm

The study examines the performance of the template-matching algorithm using a dataset derived from a simulation platform. The dataset is segregated into normal and faulty data, and ten valid segments are selected from the normal category to initialize the templates. The resulting template intervals are illustrated in Figure 14.

The residual segments are subjected to the feature extraction and comparison algorithm presented in the second section, which yields an initial classification result of the data. The outcome of the classification is tabulated in Table 2.

After calculation, the final diagnostic accuracy is 93.3%. The false alarm rate (the total number of normal segments wrongly judged as abnormal/the total number of normal segments) is 4%, and the missed detection rate (the total number of anomalous segments wrongly judged as normal/the total number of anomalous segments) is 6.9%.

Considering the practical application process, a device may receive more than 20 sound segments per day. Therefore, the occasionally missed detection can be acceptable. Nonetheless, false alarms may lead to unnecessary labor costs. As a result, in the actual application, a wide tolerance range is established in the template area to minimize false alarms while keeping the missed detection rate as low as possible.

### 4.3. Results Based on CNN

The study conducted in this experiment utilizes the CNN model to analyze a dataset obtained from a simulation platform. The final outcome of this experiment is demonstrated in Figure 15. The loss function reached a convergence point after the 50th iteration with the inclusion of 350 training samples. The tested accuracy results from the confusion matrix display exceptional classification achievements, attaining 98.4%, which indicates that the classification task is completed impeccably.

## 5. Discussion

1. Obtaining well-annotated data in real-world environments is challenging, causing our work to rely on data gathered only from a simulation platform, which has differences from the actual environment and mechanical structure of the bridge expansion joint. Despite having performed well on simulated data, the two-level classification algorithm may require further modifications to be applicable to actual scenarios based on environmental factors.

To address the above concerns, the first-level algorithm proposed in this paper is adaptive and can continuously update the template based on environmental changes, which has strong robustness. Moreover, we have currently deployed our system to collect data from some actual bridge expansion joints. Additionally, in future research, we will use these data to construct datasets to verify and improve the algorithm.

2. The second-level algorithm utilizes CNN, which requires a significant amount of data. While this approach performs well on simulated datasets, its judgment capacity becomes limited in the absence of sufficient data. Meanwhile, research in other data-driven algorithms (such as machine learning, deep learning, and optimization) remains insufficient. One potential area of future study is to establish a complete dataset and employ optimization algorithms such as GWO to optimize model parameters. However, this approach still lacks flexibility and adaptability. Alternatively, another research direction is to develop an unsupervised fault diagnosis model capable of adapting to complex factors such as environmental shifts and equipment aging, which could impact the bridge expansion joint and the judgment effect of algorithms.

## 6. Conclusions

In this paper, we have developed an IoT system to meet the periodical monitoring requirements of the health status of bridge expansion joints. The system deploys a progressive two-level classification algorithm for data processing. Specifically, edge devices perform data preprocessing and template matching, while the cloud server handles further data processing for noise reduction and CNN precise diagnosis. Additionally, we have carried out testing using the Expansion Joint Damage Simulation Data Collection Platform and developed a dataset. Based on this dataset, the first-level algorithm has achieved an accuracy rate of 93.3%, and the second-level algorithm has achieved a 98.4% accuracy rate.

The system has the following advantages:Low power consumption—The system enables the periodic switching of edge devices between normal operation and low power consumption sleep by AT commands, the energy consumption is only 2 mAh of power during a single operation, and the current in low-power mode is only a few microamps;Low computing power demand—A progressive two-level fault diagnostic algorithm can allocate computing power between edge devices and cloud servers in a reasonable manner.High accuracy—The result shows that the algorithm proposed in this paper can accurately identify faults in expansion joints.Robustness—The system can automatically update parameters and fault diagnosis templates regularly to adjust to environmental changes.

## Figures and Tables

**Figure 1 sensors-23-05090-f001:**
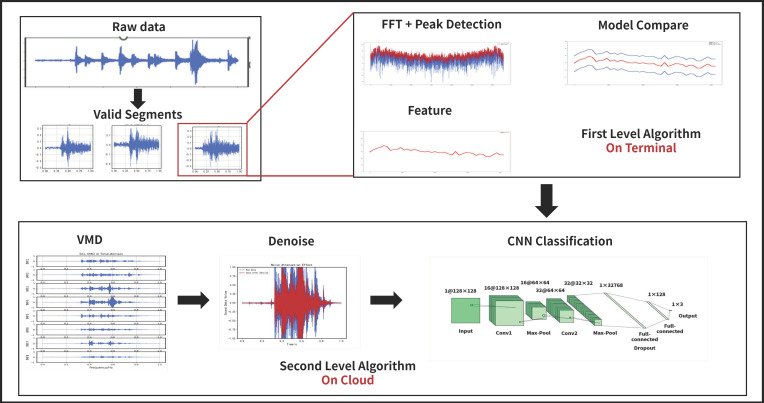
Algorithm overview.

**Figure 2 sensors-23-05090-f002:**
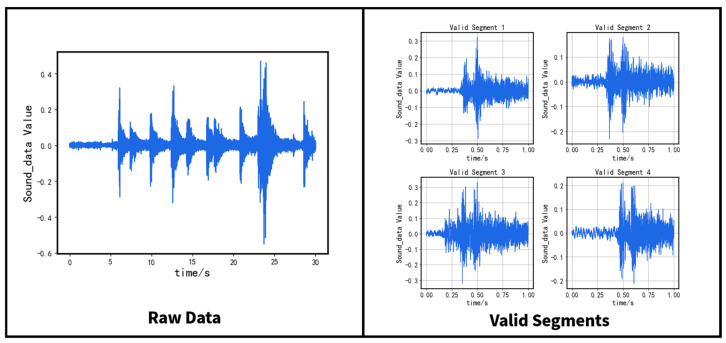
Display of slicing algorithm.

**Figure 3 sensors-23-05090-f003:**
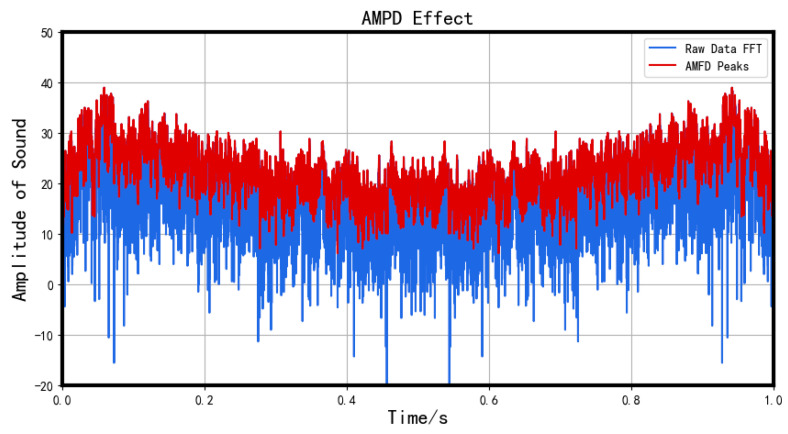
Effect of AMPD.

**Figure 4 sensors-23-05090-f004:**
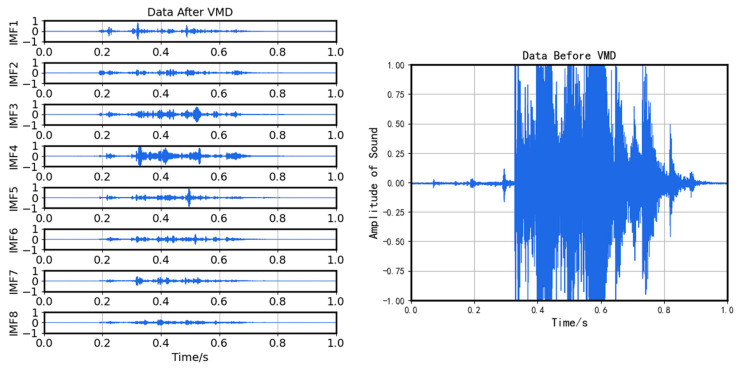
VMD results.

**Figure 5 sensors-23-05090-f005:**
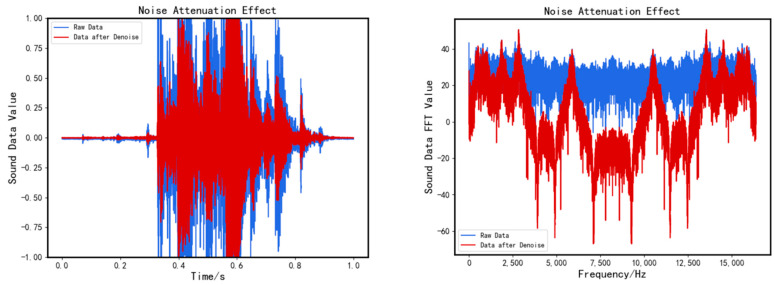
Comparison of denoise signal and original signal.

**Figure 6 sensors-23-05090-f006:**
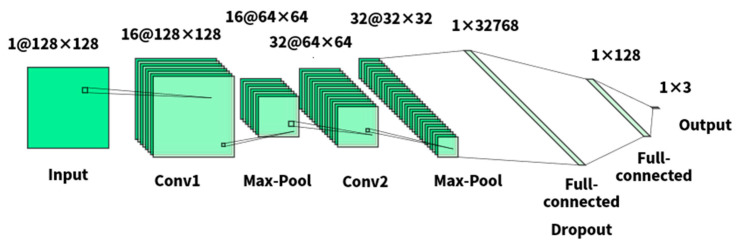
CNN model.

**Figure 7 sensors-23-05090-f007:**
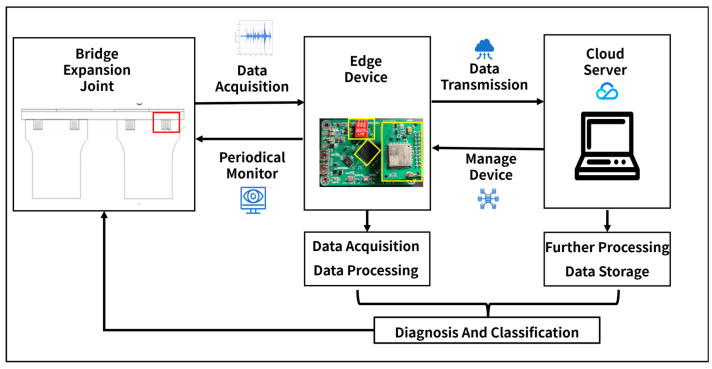
System overview.

**Figure 8 sensors-23-05090-f008:**
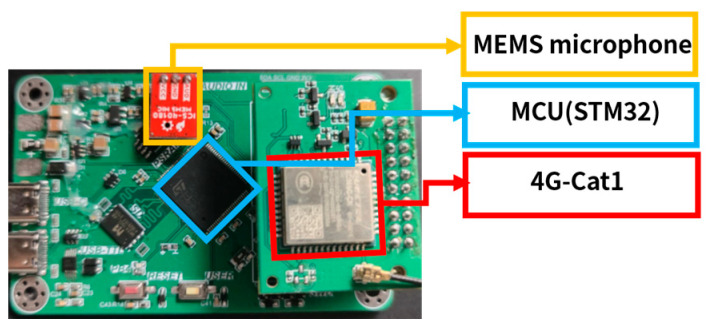
Edge data collection and processing device.

**Figure 9 sensors-23-05090-f009:**
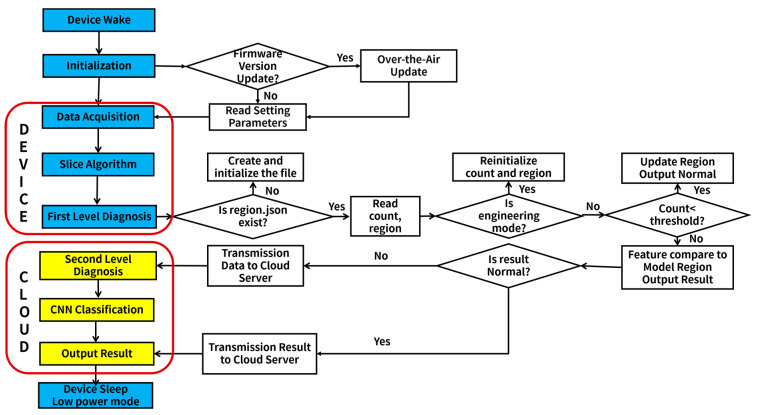
System operation process.

**Figure 10 sensors-23-05090-f010:**
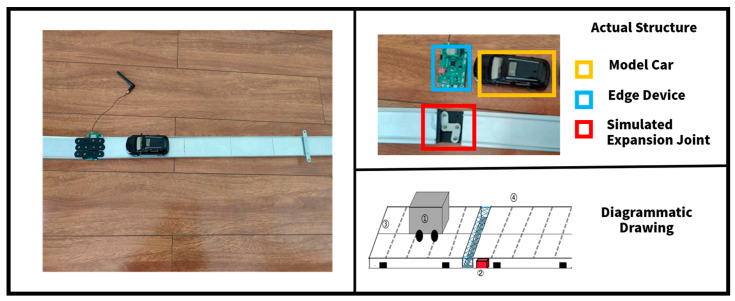
Data collection platform. In diagrammatic drawing, the white part represents the main body of bridge, the grey part is model car, the red part is edge device, the blue part is simulated expansion joint.

**Figure 11 sensors-23-05090-f011:**
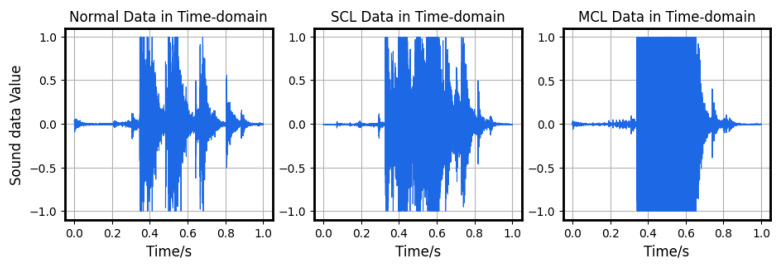
Comparison in the time-domain.

**Figure 12 sensors-23-05090-f012:**
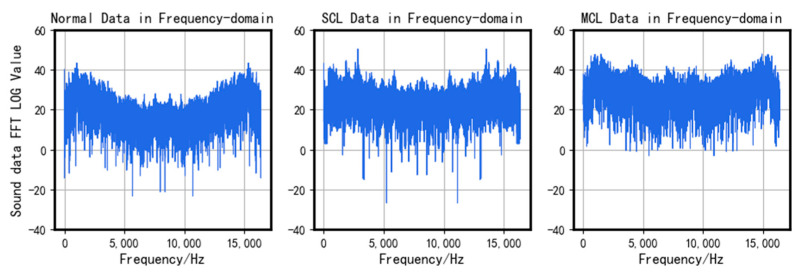
Comparison in the frequency-domain.

**Figure 13 sensors-23-05090-f013:**
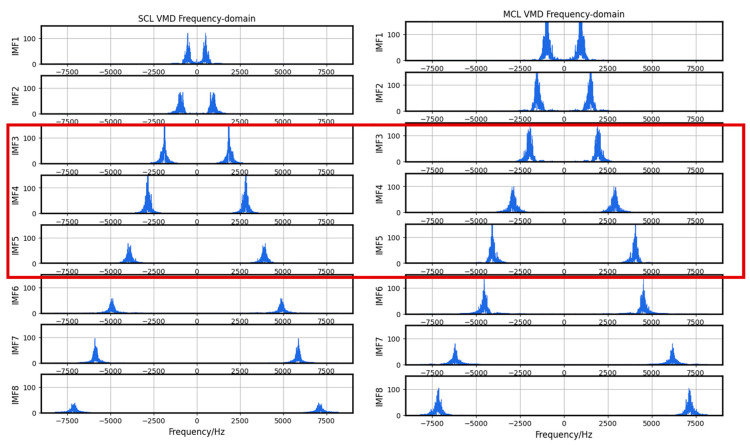
Comparison of fault data after VMD in frequency-domain. The red box frames the differences in characteristic frequencies between SCL and MCL.

**Figure 14 sensors-23-05090-f014:**
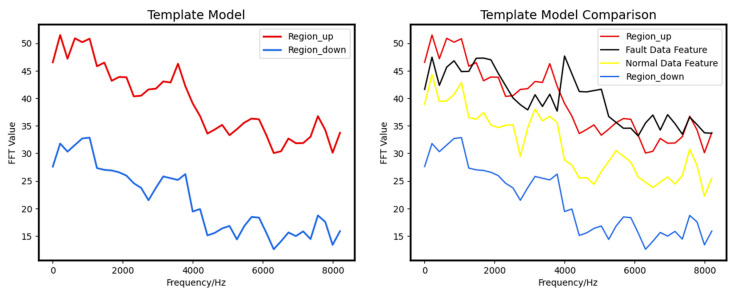
Template Matching Algorithm, the left figure is the template, and the right figure shows the comparison between the data feature and the template.

**Figure 15 sensors-23-05090-f015:**
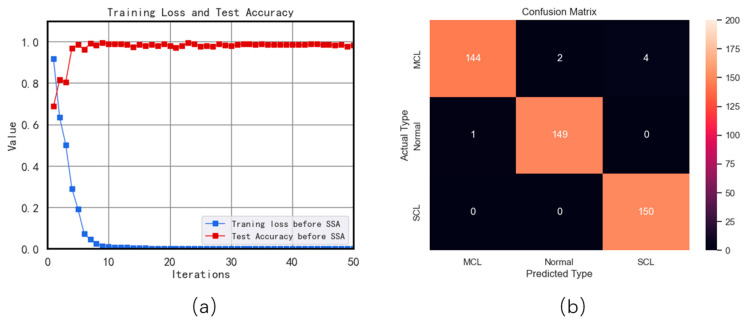
(**a**) Training loss and testing accuracy curve; (**b**) Confusion matrix based on results from CNN.

**Table 1 sensors-23-05090-t001:** Composition of the dataset.

Data Type	Number
Normal	500
Main Connect Loose	500
Subordinate Connect Loose	500

**Table 2 sensors-23-05090-t002:** Outcome of the first-level classification.

Data Real Type	Total Quantity	The Quantity Judged as Normal	The Quantity Judged as Fault or Suspected
Normal	490	460	20
Main Connect Loose	500	31	469
Subordinate Connect Loose	500	38	462

## Data Availability

Not applicable.

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
