# Peer review of "Progressive Classifier Mechanism for Bridge Expansion Joint Health Status Monitoring System Based on Acoustic Sensors"

_sensors, 2023, doi:10.3390/s23115090_

Round 1
Reviewer 1 Report
The authors present a progressive Classifier Mechanism for Bridge Expansion Joint Health Status Monitoring System Based on Acoustic Sensors. Before consider it for publication the following remarks should be considered:
• The abstract should be revised including the novelties more details.
• English typos and grammatical errors must be revised.
• Some parameters are missed to identify.
• The resolution of some Figs should be improved
• How did you consider signals as input in the CNN model
• The authors are required to add more details of data collection, input , and output in the revised manuscript.
• How can consider acoustic Emission for large scale structures ?
• The introduction should be improved including different techniques based on local and da-ta based techniques and show the accuracy of both.
• Some relevant works related to data based should be considered in the revised version such us: Multi-parameter identification of concrete dam using polynomial chaos expansion and slime mould algorithm; Damage assessment of suspension footbridge using vibration measurement data combined with a hybrid bee-genetic algorithm ; A robust FRF damage indicator combined with optimization techniques for damage assessment in complex truss structures ; Vibration-based crack prediction on a beam model using hybrid butterfly optimization algorithm with artificial neural network; Damage assessment in structures using artificial neural network working and a hybrid stochastic optimization
• The conclusion should be revised including the limitation and future work of present-ed technique.
The authors present a progressive Classifier Mechanism for Bridge Expansion Joint Health Status Monitoring System Based on Acoustic Sensors. Before consider it for publication the following remarks should be considered:
• The abstract should be revised including the novelties more details.
• English typos and grammatical errors must be revised.
• Some parameters are missed to identify.
• The resolution of some Figs should be improved
• How did you consider signals as input in the CNN model
• The authors are required to add more details of data collection, input , and output in the revised manuscript.
• How can consider acoustic Emission for large scale structures ?
• The introduction should be improved including different techniques based on local and da-ta based techniques and show the accuracy of both.
• Some relevant works related to data based should be considered in the revised version such us: An efficient artificial neural network for damage detection in bridges and beam-like structures by improving training parameters using cuckoo search algorithm ; Efficient Artificial neural networks based on a hybrid metaheuristic optimization algorithm for dam-age detection in laminated composite structures ; YUKI Algorithm and POD-RBF for Elasto-static and dynamic crack identification ; Vibration-based crack prediction on a beam model using hybrid butterfly optimization algorithm with artificial neural network ; Efficient Artifi-cial neural networks based on a hybrid metaheuristic optimization algorithm for damage detection in laminated composite structures.
• The conclusion should be revised including the limitation and future work of present-ed technique.
Reviewer 2 Report
The paper
“Progressive Classifier Mechanism for Bridge Expansion Joint Health Status Monitoring System Based on Acoustic Sensors”,
By Xulong Zhang et al.,
focuses on the use of IoT technology to improve the efficiency of bridge expansion joint maintenance. The paper proposes a low-power, high-efficiency monitoring system that uses acoustic signals to detect faults in expansion joints.
The paper proposes a two-level classifier mechanism that combines template matching and deep learning algorithms based on VMD denoising.
Overall, the proposed system seems to show promise in enhancing the efficiency of bridge expansion joint maintenance. Thus, it is an interesting reading, in line with the aim of the journal.
Nevertheless, some content and presentation issues need to be addressed before acceptance. Specifically:
1. Since authentic data related to bridge expansion joint failures are scarce, the paper establishes a simulation data collection platform to create well-annotated datasets. On the one hand, this is reasonable; yet, on the other hand, this means that the proposed procedure is only validated on data from a scaled, simplified laboratory model and lacks actual validation on real-life, real-size case studies.
2. According to the Authors, the first-level edge-end template matching algorithm achieved a 93.3% fault detection rate, while the second-level cloud-based deep learning algorithm achieved a classification accuracy of 99.0%. However, the training set and test set were divided in a ratio of 7:3, which is quite unbalanced on the training set side. Hence, these results may be affected by overtraining.
3. About the VMD denoising: in the Reviewer’s opinion, the choice of VMD is indeed a very well-made decision. However, it should be better motivated. For instance, in https://doi.org/10.3390/s21051825, it was proved that VMD outperforms other common signal decomposition/denoising alternatives such as CEEMDAN and HVD for SHM-related tasks.
4. Related to the previous comment, as with many other data-adaptive decomposition/denoising algorithms, VMD is known to perform efficiently on structures and mechanisms with well-separated, clearly visible modes, while struggling with mixed or closely-spaced modes as well as weakly excited ones. It is not clear if the proposed numerical benchmark falls into any of these compelling categories.
5. Equation 9, the meaning of the operator should be explicitly stated
6. The edge computing system is claimed to be ‘low-power’ and ‘high-efficiency’. However, it would be better to quantify these statements and, most importantly, compare them to what is generally expected to be the average power consumption/efficiency of state-of-the-art, already-existing alternatives.
7. In the Introduction, the state-of-the-art review for bridge monitoring (not necessarily restricted to bridge expansion joints only) is a bit limited. There are many recent studies, e.g. https://doi.org/10.1016/j.engstruct.2022.115573 and related ones, that could be reported to provide some further context.
8. Figure 7: rather than ‘Data Processing’ and ‘Further Processing’, wouldn’t be better to say ‘Data Pre-Processing’ and ‘Data Processing’?
Overall, the English of the paper is good, and its structure is clear and easy to follow. Nevertheless, throughout the whole manuscript, some editorial issues should be fixed (e.g. line 71, Torres M. E. et al should not be all capitalised; line 119, the font of the title 'Overview' is different from the rest of the text).
Reviewer 3 Report
Comments on the paper submitted to Sensors (ISSN 1424-8220)
Title: Progressive Classifier Mechanism for Bridge Expansion Joint Health Status Monitoring System Based on Acoustic Sensors
Manuscript ID: sensors-2400369
Conclusion of my review report: The paper presents a well-structured and comprehensive study. The contribution of the work to the field is high, and the paper is well written, in general. However, I missed some Figures related to the methods, which would be important for future applications. But in general, these corrections can be easily done.
Because of this, my final recommendation is to accept after minor revisions.
Title: The title seems adequate.
Abstract: The abstract is well structured since it provides the purpose, methods and main results. In general, I only missed the statement of a final conclusion on the work. It seems that the abstract finishes with the "results" and not with the conclusion.
Technical / Editorial comments / Suggestions/Questions to the authors
Line 10 – the term “IoT” shall be defined before the use of the abbreviation. I imagine that refers to “Internet of Things”
Line 16 – Please, consider giving the meaning of "VDM" before the use of the abbreviation. “Variational Mode Decomposition”.
Line 19 – Please, consider adding a final conclusion to the abstract after the results.
Section 1: Introduction
General: The authors present the main ideas required in the Introduction adequately: (i) context, (ii) motivation of the proposed research; (iii) purpose of the paper and (iv) methods applied.
The literature review is comprehensive and supports the motivation and methods applied.
I have only some minor corrections that need to be addressed.
Technical / Editorial comments / Suggestions/Questions to the authors
Line 49 – 50: “Presently, a significant amount of research has been devoted to applying vibration signals in the development of structural health monitoring algorithms.” Please, consider adding references to sustain this argument.
Lines 80-82: It seems contradictory motivation. Please, consider rewriting this sentence.
Line 71 – Correct the citation of “Torres”.
Sections 2: Progressive Two-Level Fault Diagnosis Algorithm
The authors provided adequate background information about the methods applied: (i) Slice (extract valid segments); (ii) Automatic Peak Detection(AMPD); (iii) Template Matching-Based Determination Algorithm (First Level) and (iv) CNN classifier based on VMD denoising (Second Level)
In general, the methods were well illustrated by Figures.
Technical / Editorial comments / Suggestions/Questions to the authors
Figure 2 – The terms FFT, CNN and AMPD were not previously defined. Consider adding this explanation.
Line 214- The term “IMF” was not previously defined in the text.
3. Periodical Monitoring System
General: This section is well structured and provides important information about the system's operation.
I have only some minor comments.
Technical / Editorial comments / Suggestions/Questions to the authors
Line 310: MEMS microphones are installed in edge devices under the expansion joints of the bridge”. Please, consider adding Figures from such equipment installed at the bridge.
Line 321-328 – Please, consider adding Figures from such equipments with the corresponding identification.
Line 331: The description and explanation from Figure 8 should be enhanced.
Figure 9: the size of the parts of the reduced-scale experiments should be given in more detail.
Section 4: Results:
The results are well presented and supported by the presented methods. However, I identified some aspects that need to be improved.
Technical / Editorial comments / Suggestions/Questions to the authors
Line 390 - 394: The description would be better if the author added Figures sketching each scenario tested: normal, main connection loose (MCL), and Subordinate connection plate loose (SCL).
Line 427 – 430: Are the percentages correct? Should the total result in 100% ?
Section 5: Conclusions
The conclusions are supported by the presented methods and results. Besides, they are valuable for future applications.
English writing:
In general, the manuscript was written in good English.
Figures and tables.
Most Figures allow an understanding of the methods and results. But I point out some places where I missed Figures for a better understanding of some methods. I also identified some places on which the Figures should be improved in detail.
In general, the manuscript was written in good English.
Round 2
Reviewer 2 Report
This Reviewer is satisfied with the replies provided by the Authors and the changes made to the manuscript.
The current version of the paper can be accepted after final grammar- and proof-checking.